# Comparison between Piezoelectric Filter and Passive LC Filter in a Class L−Piezo Inverter

**Vincent Massavie** [1,*] **, Ghislain Despesse** [1] **, Sebastien Carcouet** [1] **and Xavier Maynard** [2]

1 University Grenoble Alpes, CEA, Leti, F-38000 Grenoble, France; ghislain.despesse@cea.fr (G.D.); sebastien.carcouet@cea.fr (S.C.)
2 University Grenoble Alpes, CEA, Liten, F-38000 Grenoble, France; xavier.maynard@cea.fr
* Correspondence: vincent.massavie@cea.fr

**Abstract:** This paper presents a comparison between piezoelectric filtering and passive LC filtering integrated into an HF class L−Piezo inverter. This L−Piezo inverter is a variant of class φ2 where the filtering of the second harmonic is carried out by a piezoelectric resonator. Piezoelectric filters are well known in the signal domain (RF filtering), but their use in the field of power electronics, as a temporary energy storage element, is rather recent. In power electronics, piezoelectricity has mainly been used as a transformer, in particular, to greatly increase voltages (backlight applications). A class L−Piezo inverter with Lithium Niobate (LNO) piezoelectric resonator is designed for a switching frequency of 10.4 MHz, an input voltage of 30 V, and an output power of 15 W. To compare these two filtering methods, two prototypes are built, one with piezoelectric filtering and one with passive LC filtering. Measurements show a reduction of 60% of the losses in the filter, while the volume of the filter is reduced by a factor of 50.

**Keywords:** class φ2; class L−Piezo; HF−VHF inverter; passive LC filter; piezoelectric resonator; power electronics; resonant inverter

## 1. Introduction

The development of semiconductor technologies since the middle of the 20th century has made the production of increasingly efficient and compact power supplies possible. So much so that power electronics has become omnipresent in our current society. At constant transmitted power, increasing the switching frequency reduces the energy to be stored, at each period, in passive elements, namely capacitors and inductors. The latter have, therefore, a lower value and a reduced size. VHF (Very High Frequency) conversion meets the growing need for the miniaturization of electronic equipment by reducing the size of its passive components. VHF converters are converters that use a resonant circuit to transmit power. The class E inverter [1–3] is the most used in the industry because it is easy to size and requires few components. However, the voltage stress on the switch is high (3–4 times the input voltage), inducing an oversizing of the active component and requiring the use of a choke inductor (high−value inductor) at the input [1–3]. To improve this last point, the class E2 inverter [3,4] reproduces class E waveforms by using a resonant circuit tuned to the second harmonic of the switch voltage. This resonant circuit allows the use of a lower value input inductor, reducing the size and weight of the latter, but this topology does not solve the problem of voltage stresses on the switch equal to 3–4 times the input voltage. The φ2 class inverter is a VHF inverter topology that is more complex in size than the two others mentioned above but which offers the advantage of reducing the voltage stresses on the switch to twice the input voltage [3,5]. The disadvantage of these different topologies lies in their inductors. Indeed, inductors using magnetic materials have a quality factor that decreases with frequency [6,7]. Beyond ten MHz, using an air core inductance devoid of magnetic materials becomes mandatory to maintain a good quality

factor and not degrade the efficiency. However, these air core inductors are often bulky and conflict with the miniaturization objective mentioned for the frequency increase.

To overcome this problem, the use of piezoelectric resonators has been studied. Piezo-electric components offer several advantages, such as a power density and quality factor much higher than inductors, a planar form factor, and a reduced electromagnetic field. Piezoelectric materials are used in many fields, particularly as sensors or actuators [8–11]. Recently, the research seeks to integrate piezoelectric components into power electronics. As shown in [12–14], a piezoelectric resonator is used to replace the magnetic energy storage of an inductor with mechanical storage. The use of piezoelectricity as an energy storage element in power electronics has been made possible by the significant improvement in the electromechanical coupling coefficients of piezoelectric materials in recent years. Today, some electromechanical coupling factors exceed 0.5, reducing the mechanical energy level stored in the material compared to the electrical one exchanged at each cycle, making a large power density possible. In terms of efficiency, the advantage of limited stored mechanical energy and a high−quality factor allows high efficiency in power exchange through piezoelectric storage [15,16].

In this paper, we study a piezoelectric resonator used as a power filter in a VHF converter topology. In Section 2, the working principle and the design of the class L−Piezo are developed. Section 3 shows a theoretical approach based on a characterized piezoelectric resonator in Lithium Niobate and a simulation of the class L−Piezo inverter. Section 4 compares the experimental results of a class L−Piezo inverter using a piezoelectric filter and a prototype using passive components $L_m − C_m − C_0$ for power filtering. A conclusion summarizes the work in Section 5.

## 2. Class L−Piezo Inverter

### 2.1. Operating Principle and Design

The principle of this inverter is the same as the class φ2 inverter [17], namely filtering the second harmonic to reduce the voltage peaks on the transistor. To cancel the second harmonic, the class L−Piezo [18,19] uses a piezoelectric resonator that vibrates at twice the operating frequency of the inverter. A piezoelectric resonator allows great stability on the filtering frequency, a high−quality factor compared to a classic LC branch, and a slim and compact size. The class L−Piezo inverter is depicted in Figure 1a.

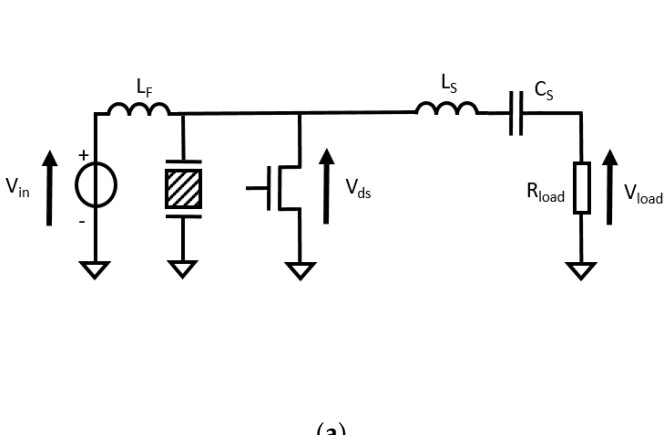
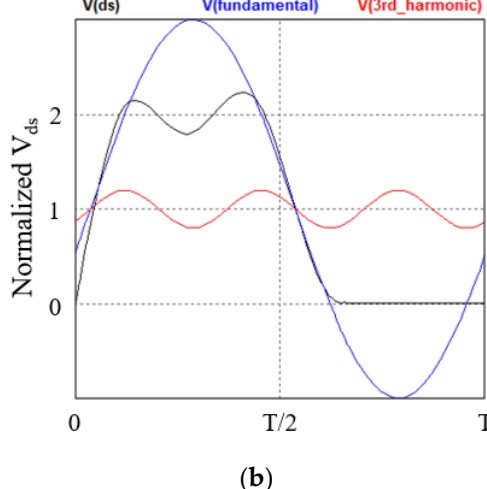

(**a**)          (**b**)

**Figure 1.** Schematic of an L−Piezo class inverter (**a**), and typical drain voltage ($V_{ds}$) of class L−Piezo inverter and its harmonics (**b**).

A normalized drain voltage of the class LPiezo inverter (in black) is shown in Figure 1b with its fundamental (in blue) and its third harmonic (in red). The impedance of $L_F$ in parallel with the piezoelectric is sized to have a high impedance at the switching frequency at

three times the switching frequency and a low impedance at twice the switching frequency. Finally, the on/off switching induces a $V_{ds}$ signal composed mainly of a fundamental and a third harmonic, as shown in Figure 1b. The second harmonic is canceled by the piezoelectric resonance. This association of the fundamental and the third harmonic with the cancelation of the second harmonic decreases the maximum voltage $V_{ds}$.

### 2.2. Zero Voltage Switching (ZVS)

If $V_{ds}$ go down to zero before the switch is turned on, a reverse voltage and reverse losses appear, as the quasi−ZVS shows in Figure 2a. On the opposite, if the switch turns on before $V_{ds}$ reaches zero, it is the hard switching shown in Figure 2b. In both cases, switching losses decrease power efficiency.

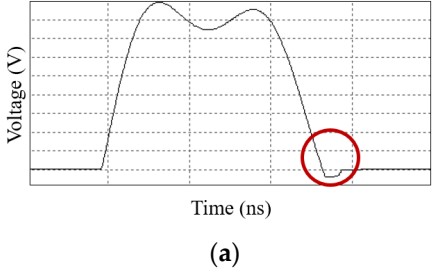

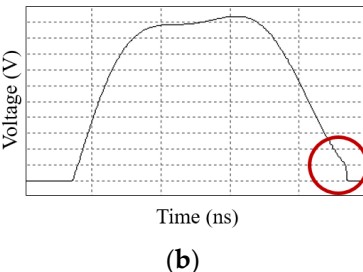

(**a**)                                                                                                  (**b**)

**Figure 2.** Drain−source voltage waveform in (**a**) quasi−ZVS and in (**b**) hard switching.

In [5], ZVS is achieved when the input impedance at the fundamental of the switching frequency is between 30° and 60° inductive and when the impedance at the third harmonic is capacitive, and its magnitude is several dB (between 4 dB and 8 dB) below the impedance magnitude at the fundamental. In our case, the input impedance can be tuned by modifying the input inductance value $L_m$.

The steps we follow here for sizing the L−Piezo class inverter are:

- Characterization of a piezoelectric resonator;
- Calculate input inductance $L_f$ with class φ2 equation;
- Tuning the inverter to reach the optimal behavior (ZVS, minimal voltage stress on the switch, etc.).

The detailed equations for sizing the L−piezo are given in [6].

### 3. Piezoelectric Model Extraction

#### 3.1. Determination of the Piezoelectric Resonator Model from Material Data and Size

For a converter that must operate at the frequency *f*, it is necessary to find a piezoelectric resonator having a resonant mode at 2*f* (because its purpose is to cancel the harmonic 2).

A piezoelectric resonator operating around one of its resonant frequencies can be modeled by an RLC circuit in parallel with a capacitor, as shown in Figure 3 (Butterworth Van Dyke model [20]).

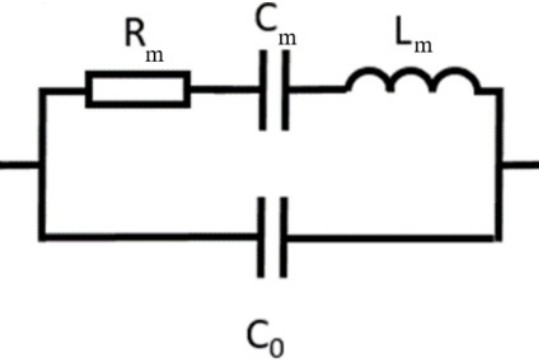

**Figure 3.** The Butterworth Van−Dyke equivalent circuit model.

The RLC branch models the mechanical behavior of the material. $C_0$ is the electrical capacitance of the material (2 electrodes separated by an insulator).

As we focus on high−frequency operation (>5 MHz), we only consider piezoelectric material operation in thickness mode (wavelength < 1 mm). The first resonant mode frequency of the resonator in thickness mode can be determined by the following formula:

$$f_r = \frac{N_t}{t} \tag{1}$$

where $N_t$ is the frequency constant of the material in thickness mode and $t$ is the thickness of the material.

The blocked capacitance $C_0$ for a piezoelectric disc is:

$$C_0 = \varepsilon_r.\varepsilon_0.\frac{A}{t} \tag{2}$$

where $\varepsilon_0$ is vacuum permittivity ($\varepsilon_0$ = 8.854 × 10 − 12 F/m), $\varepsilon_r$ is relative permittivity, $A$ is electrode area, and $t$ is inter−electrode distance.

The coupling factor $k_t$ represents the proportion of electrical energy converted into mechanical energy and vice versa in thickness mode. The motional capacitance $C_m$ is calculated from $C_0$ and $k_t$ as follows:

$$C_m = k_t^2.C_0 \tag{3}$$

By considering the resonant frequency $f_r$ as input data, $L_m$ can be calculated as:

$$L_m = \frac{1}{(2.\pi.f_r)^2.C_m} \tag{4}$$

Finally, the last component to calculate is $R_m$, representing the losses of the piezoelectric resonator. Based on the quality factor $Q$, $R_m$ can be expressed by:

$$R_m = \frac{(2.\pi.f_r).L_m}{Q} \tag{5}$$

*3.2. Lithium Niobate Material Case (LNO)*

Among the many existing piezoelectric materials, PZT and LNO have the best figure of merit $k^2Q$, high enough to be used in a power converter to exchange energy [15]. In [18], we have already shown an interest in PZT for use at 1 MHz. However, for higher frequencies, the parallel capacitance of the PZT induces a too−large current ($1/(C_0\omega)$ too low when $f$ > 5 MHz). This is not the case for LNO, which has a very low dielectric constant compared to PZT (28.7 for LNO Y−36° cut [21] vs. 1470 for PZT−C213 [22]).

The properties of LNO depend on the cutting angle relative to the crystal orientation of the material. Table 1 shows the characteristics of $Y-36°$ cut LNO (so a cutting angle of $36°$ on the $Y-$axis relative to the orientation of the crystal) and considers a resonance in thickness mode.

**Table 1.** Characteristics of $Y-36°$ cut LNO resonant in thickness mode.

|  | LNO |
|---|---|
| Frequency constant (MHz/m) | 3300 |
| Dielectric constant | 28.7 |
| Coupling factor k | 0.49 |
| Electrical quality factor $Q_e$ | 1000 |
| Mechanical quality factor $Q_m$ | $1.5 \times 107$ à 6.3 MHz [23] |
| Curie temperature | 1145 °C |

Thanks to its high figure of merit $k^2 Q = 240$, this material promises a low level of losses, as expressed in [17]. This material is available in wafer shape that can be thinned and cut.

The manufacturer's data does not make it possible to know the maximum current that can circulate in the piezoelectric resonator; however, this information is an important dimensioning criterion in this application. We, therefore, propose to estimate it in the next section.

### 3.3. Current Density and Pre$-$Sizing Equations

The piezoelectric resonator has a mechanical strength limit and a thermal limit inducing a maximum current density limit of $J_{max}$ in its motional branch. This maximum current density depends on the oscillation frequency of the resonator.

This maximum current density is determined experimentally. By experiment feedback on the LNO [14,15], we observe an increase in the maximum current density by a factor of five per decade of frequency. For $f_1 = 6$ MHz, the maximum current density of LNO is estimated at $J_{max} = 1$ A$_{crête}$/cm$^2$ [15,24].

$$J_{max}(f) = J_{max}(f_1)\left(\frac{f}{f_1}\right)^{\frac{ln(5)}{ln(10)}} \tag{6}$$

Let us take a factor of 10 in frequency increase as an example. Considering the thickness mode, this increase is equivalent to dividing the thickness by 10, as shown in Equation (1).

The external capacitance $C_0$ of the piezoelectric resonator per unit area is multiplied by ten, cf. Equation (7).

$$C_{0/cm^2} = \frac{\varepsilon_0\varepsilon_r.1cm^2}{e} = \frac{\varepsilon_0\varepsilon_r.1cm^2}{N_t}f \tag{7}$$

The maximum open circuit achievable voltage by the resonator is defined by Equation (8), where all the current density capability is just used to charge and discharge $C_0$ (circulating current).

$$V_{max}(f) = \frac{J_{max}}{2\pi f C_{0/cm^2}} = \frac{J_{max}(f_1)\left(\frac{f}{f_1}\right)^{\frac{ln(5)}{ln(10)}}}{\frac{\varepsilon_0\varepsilon_r.1cm^2}{N_t}f} \tag{8}$$

Figure 4 shows the maximum allowable voltage by a piezoelectric resonator in $Y-36°$ cut LNO for frequencies from 1 MHz to 60 MHz.

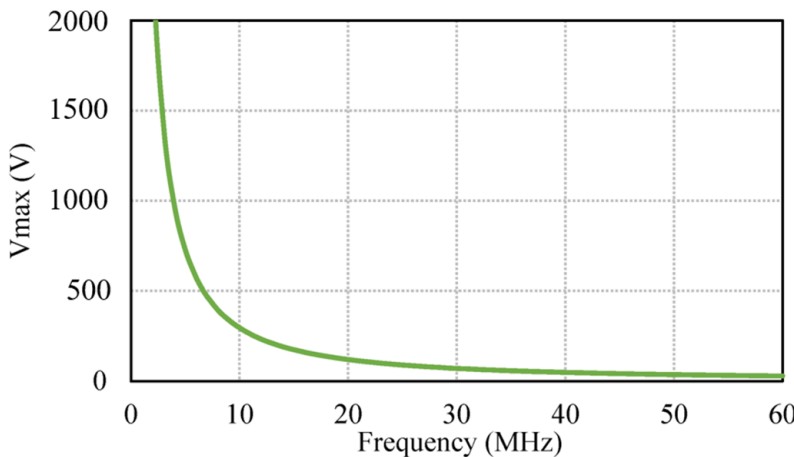

**Figure 4.** Maximum allowable voltage by the piezoelectric resonator as a function of the frequency for Y−36° cut LNO.

Knowing that the current density $J_{max}$ increases by a factor of five per decade of frequency and the impedance $1/(C_0\omega)$ decreases by a factor of 100 per decade, increasing the frequency by a factor of 10 and decreasing the maximum achievable voltage by a factor of twenty.

It shows that a piezoelectric resonator admits a frequency limit depending on the voltage amplitude seen by the resonator. This is an upper limit; it does not consider the current exchanged with the outside.

For sizing the LNO surface, the amplitude of the voltage seen by the resonator in the converter is first defined. In the case of an L−Piezo inverter, the resonator sees the drain−source voltage of the transistor, which is around twice the input voltage.

We can deduce the circulating current in the resonator:

$$I_{circulating} = 4\pi f C_{0/\text{cm}^2} V_{in} \tag{9}$$

The available current that can be exchanged with the outside is written:

$$I_{available/\text{cm}^2} = J_{max}(f) - I_{circulating} = J_{max}(f) - 4\pi f C_{0/\text{cm}^2} V_{in} \tag{10}$$

Figure 5 shows the available current under 20 V for frequencies from 1 MHz to 60 MHz.

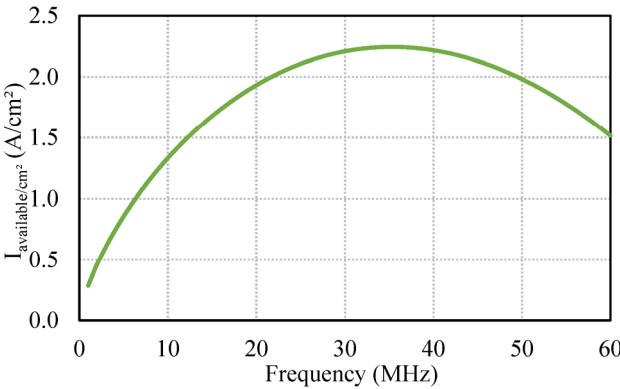

**Figure 5.** Available current per cm² under 20 V depending on the operating frequency for LNO.

The useful current that the piezoelectric resonator needs to exchange with the outside is determined by simulation, which gives a minimum useful current of $I_{u\_min}$.

The minimum surface of the piezoelectric resonator is then given by:

$$S_{min} = \frac{I_{u\_min}}{I_{available}/\mathrm{cm}^2} \tag{11}$$

Figure 6 shows the minimum LNO area required to provide a useful current of 1 A under 20 V.

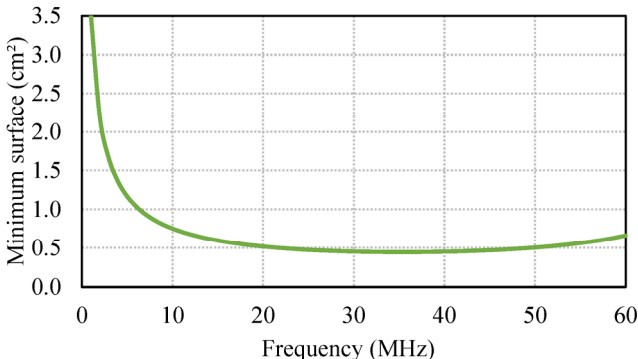

**Figure 6.** Minimum LNO area to provide 1 A of useful current under 20 V according to the frequency.

For 1 A of useful current and under 20 V of voltage excursion, the minimum surface of the resonator required is less than 1 cm$^2$ for a frequency greater than 6.1 MHz.

## 4. Simulation of Class L−Piezo Inverter

In this part, we use a piezoelectric resonator in LNO to design an L−Piezo inverter. We start by characterizing the piezoelectric resonator, and then we determine the characteristics of the inverter according to the voltage and current limits of the resonator.

### 4.1. Piezoelectric Resonator Characterisation

The converter is built around a piezoelectric resonator. We use an LNO resonator with a thickness of 150 μm for 1 cm$^2$ of useful surface. The other components of the converter will be sized according to the characteristics of this resonator.

From these dimensions, we determine the resonant frequency in the thickness mode of the resonator thanks to Equation (1) with $N_t$ = 3.3 MHz.mm for LNO [23].

The resonance frequency in thickness mode is 20.8 MHz.

The equivalent model can be deduced using the electromechanical characteristics of the Y−36° cut LNO and the equations seen in part 2. Values of the equivalent model are presented in Table 2.

**Table 2.** Values of the model of the piezoelectric resonator.

| Model | Value |
|:-----:|:-----:|
| $C_0$ | 180 pF |
| $L_m$ | 1.3 μH |
| $C_m$ | 45 pF |
| $R_m$ | 78 mΩ |

### 4.2. Design of a 10.4 MHz Class L−Piezo Inverter

By taking equations of part III.B, the maximum current and the maximum voltage admissible by the resonator are calculated.

For a resonance at 20.8 MHz, we have the following:

- A maximum current of 2.48 A;
- A maximum voltage of 105.9 V.

To avoid reaching the voltage limit of the resonator, we size the inverter for an input voltage of 30 V (60 V maximum at the terminals of the resonator).

The available current that the resonator can exchange with the inverter is calculated using Equation (10) and is 1 A at 20.8 MHz and 60 $V_{peak\_to\_peak}$.

With 1 A and 30 V input voltage, by simulation, a converter of 30 W is reachable. To avoid placing the LNO at its extreme operating limit, the converter is sized for 15 W.

The resonant frequency of the piezoelectric resonator is 20.8 MHz. The switching frequency of the L−Piezo is chosen at 20.8 MHz/2 = 10.4 MHz to filter the second harmonic of the drain−source voltage.

With these characteristics and the equations of part 2, we can calculate the values of the other components of the inverter, gathered in Table 3.

Once sized, the inverter is simulated on LTspice; the simulation circuit is given in Figure 7a. The main voltages waveforms of the class L−Piezo inverter are shown in Figure 7b.

Figure 7b shows that the voltage $V_{ds}$ has a maximum of 63 V, corresponding to around twice the input voltage $V_{in}$. Just before the switch closes, the voltage $V_{ds}$ is zero, so the ZVS is reached. The amplitude of the output voltage $V_{load\_peak}$ is 33 V.

The purpose of the $L_s-C_s$ output filter is to filter the fundamental of the $V_{ds}$ voltage to obtain a sinusoidal voltage at the output. However, the impedance of this filter appears in the sizing equations of the inverter. The couple of value $L_s-C_s$ must be selected in such a way as to provide the necessary reactance to obtain the desired power at the output of the inverter [5,18]. There is then a compromise to be made between the transmitted power and the waveform of $V_{load}$.

**Table 3.** Values of the components of the L−piezo converter.

| Model | Value |
|---|---|
| $L_f$ | 357 nH |
| Piezoelectric resonator model | $C_0$ = 180 pF<br>$L_m$ = 1.3 μH<br>$C_m$ = 45 pF<br>$R_m$ = 78 mΩ |
| Switch | GaN HEMT model [25] |
| $L_s$ | 520 nH |
| $C_s$ | 500 pF |
| $R_{load}$ | 50 Ω |

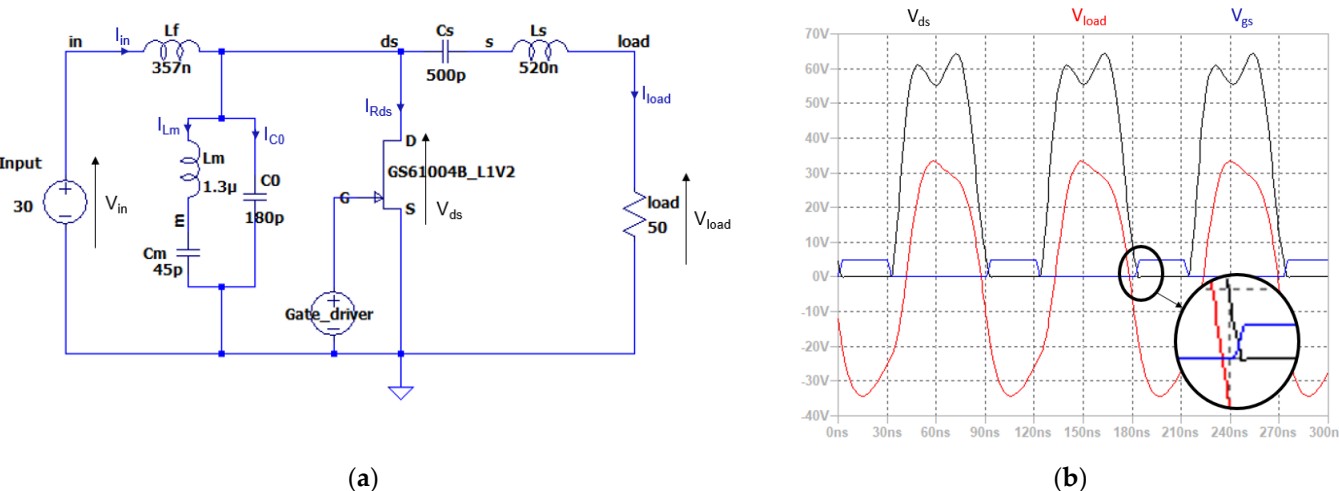

(**a**)  (**b**)

**Figure 7.** LTspice simulation model of the L−Piezo inverter (**a**), and main voltages waveforms of the class L−Piezo inverter (**b**).

## 5. Experimental Results

This part presents the experimental results obtained with two prototype inverters, one with the piezoelectric filter and one using a classic LC filter.

### 5.1. Prototype including a Piezoelectric Material

To perform an experimental validation, we have established a test bench (Figure 8a). The test equipment consisted of the following:

- A double continuous power supply (for the input voltage and to supply the driver);
- A Keysight 81,150 A wave generator (to create and send the transistor gate control signal);
- An oscilloscope;
- A 50 $\Omega$ Pasternack load.

Figure 8b shows the class L−Piezo Inverter in operation with a switching frequency of $f_s$ = 10.4 MHz, an input voltage of $V_{in}$ = 30 V, and an output power of $P_{load}$ = 15 W.

Figure 9a shows the assembled PCB of the class L−Piezo inverter described here.

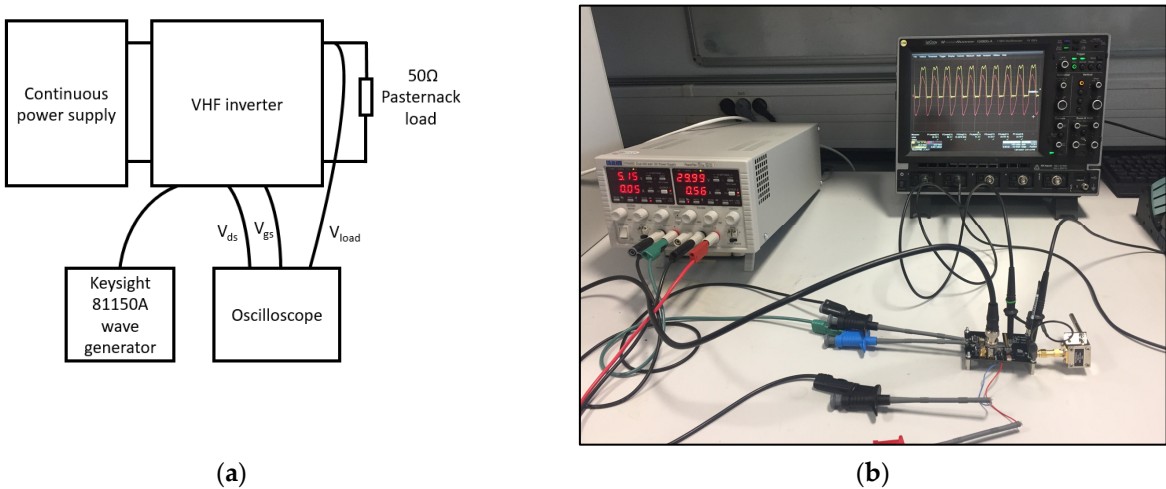

(**a**)                                                    (**b**)

**Figure 8.** Diagram of the test bench (**a**), and L−Piezo inverter in operation (**b**).

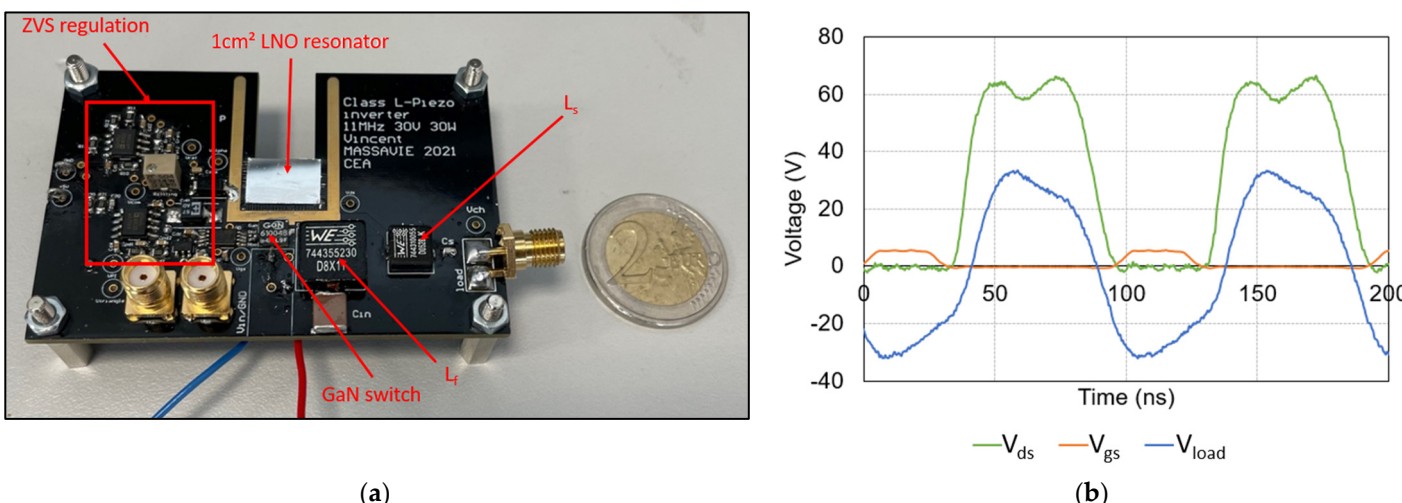

(**a**)                                                    (**b**)

**Figure 9.** Prototype with piezoelectric filter (**a**), and characteristic voltages of the L−Piezo inverter (**b**).

The regulation of ZVS (Zero Voltage Switching) that appears in Figure 9a is the subject of another article [26]. For this article, this circuit is not used; the inverter operates in an open loop with a duty cycle of 0.3.

The experimental main voltages waveforms of the L−Piezo inverter prototype are shown in Figure 9b. The maximum value of the drain voltage $V_{ds}$ is 62 V which is close to twice the input voltage $V_{in}$ = 30 V. This value agrees with the simulation. The output voltage amplitude $V_{load\_peak}$ is 33.5 V.

Waveforms conform to our theoretical expectations. The objective is now to know if piezoelectric filtering is more relevant than classic LC filtering.

### 5.2. Prototype without Piezoelectric Material

To check the relevance of using a piezoelectric filtering, we made a second piezoelectric–less prototype for comparison (Figure 10a).

This second prototype has the same characteristics as the L−Piezo inverter; the only difference is the use of a passive LC filter (discrete capacitance and inductor components) to cancel the second harmonic of the drain−source voltage.

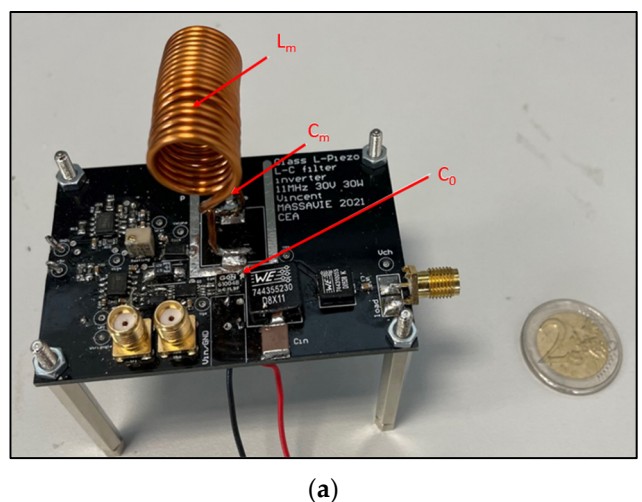

(**a**)

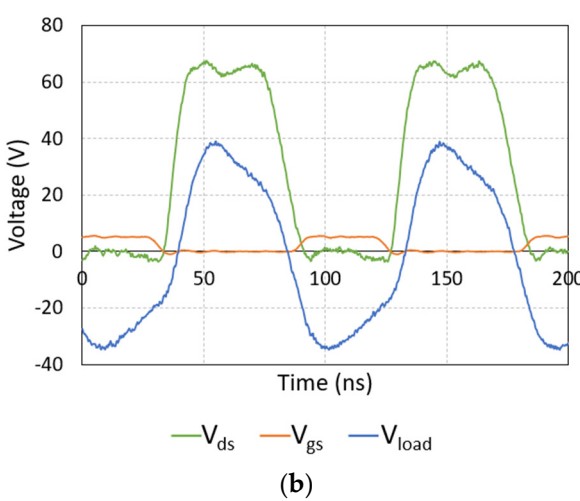

(**b**)

**Figure 10.** Prototype with passive LC filter (**a**) and characteristic voltages of the inverter (**b**).

The LC filter inductor is a hand−wound air inductor with a 0.75 mm$^2$ section copper wire. The use of an air inductance can be explained for two reasons:

- The current flowing in the filter oscillates at 20.8 MHz. At this frequency, magnetic core inductors have a low−quality factor, increasing component losses. Air core inductors are more interesting for very high frequencies. Currents through the other inductors in the inverter only oscillate at 10.4 MHz, which is still within the operating range of some magnetic core inductors;
- A hand−wound air inductor allows precise control over the actual value of the inductor.

The measured characteristics of the discrete components used for the passive LC filter are grouped in Table 4.

**Table 4.** Values of the passive components used to replace the piezoelectric resonator.

| Component | Component Characteristic | Measured Value |
|:---:|:---:|:---:|
| $C_0$ | 1 × C0603C181F1GACTU | 182 pF |
| $L_m$ | 18 turns of Ø1.56 mm copper wire with 11.88 mm/15 mm diam. | 1.394 μH |
| $C_m$ | 1 × C1206C430JBGACTU | 41.2 pF |

Figure 11 compares calculated and real impedance measured with a Keysight E5061B network analyzer for the piezoelectric filter and the passive discrete components ($L_m$ + $C_m$ + $C_0$).

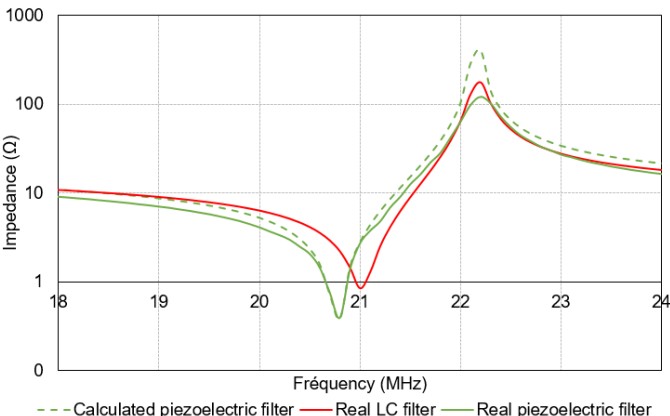

**Figure 11.** Comparison of calculated and measured impedance filters around the main thickness resonant frequency.

The theoretical model of a piezoelectric resonator is similar to the one measured. The resonant frequency of the LC filter is slightly shifted due to the tolerance on the capacitance used (tolerance of $\pm5$%). The impedance at the resonant frequency is significantly higher by using passive components, increasing the losses. This is due to the quality factor of the inductance ($Q_{Lm}$ = 203), which is lower than that of the piezoelectric resonator ($Q_{piezo}$ = 802).

*5.3. Efficiency Comparison*

An efficiency measurement was performed for various input voltages with both prototypes (piezoelectric filter and passive LC filter). The results are presented in Figure 12.

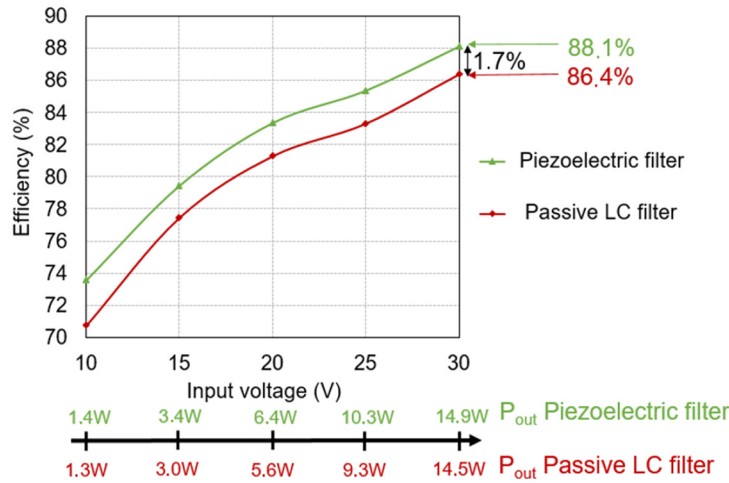

**Figure 12.** Comparison of the measured efficiency of the L−Piezo inverter with a piezoelectric filter and a passive LC filter for several input voltage values.

The measured efficiency shown in Figure 12 is a function of the input voltage; each input voltage corresponds to an output power also reported for both prototypes. $P_{in}$ is measured by measuring the DC voltage $V_{in}$ and the DC input current $I_{in}$. The output power is obtained by computing the ratio between the square of the effective voltage of $V_{load}$ measured with the oscilloscope and the value of the Pasternack load of 50 $\Omega$.

For an input voltage of $V_{in}$ = 30 V, the efficiency is 88.1% for the piezoelectric filtering compared to 86.4% for the LC filtering, i.e., a gain of 1.7% on the overall efficiency and a reduction by 12.5% of the losses. This gain is due to the high−quality factor of the piezoelectric resonator compared to that of an inductor. Note that this efficiency gain is independent of the input voltage.

### 5.4. Losses Distribution

In this part, we study more deeply the distribution of losses in the converter in simulation to identify the impact of the piezoelectric resonator.

An inductor is not ideal. The losses can be modeled by introducing resistance in series with a perfect inductor. This resistance is directly correlated to losses in the wire constituting the coil. The quality factor of the inductance is the ratio between the reactance of the coil at a given frequency and its resistance. The higher the quality factor is, the closer the inductance approaches an ideal behavior. The quality factor is given by the following equation:

$$Q_L = \frac{X_L}{R_L} = \frac{Lw}{R_L} \tag{12}$$

Similarly, for capacitors, the quality factor refers to the ratio between the capacitor's reactance and its resistance. It is given by the equation:

$$Q_C = \frac{X_C}{R_C} = \frac{1}{wCR_C} \tag{13}$$

To obtain a simulation as close as possible to the real case, an impedance measurement of each passive component of both prototypes is carried out. The impedance measurement was performed by using a Keysight E5061B network analyzer. All the following quality factors are given in Table 5 for a frequency of 10.4 MHz except for the piezoelectric resonator, the inductor $L_m$ and the capacitor $C_m$ measured at 20.8 MHz. Prototype 1 is the name of the L−Piezo inverter using piezoelectric filtering, and prototype 2 is the name of the inverter using passive LC filtering.

**Table 5.** Components quality factor of both prototypes.

| Components | Quality Factor for Prototype 1 | Quality Factor for Prototype 2 |
|:---:|:---:|:---:|
| $L_f$ | 55 | 55 |
| Piezoelectric resonator | 802 | / |
| $L_m$ | / | 203 |
| $C_m$ | / | 998 |
| $C_0$ | / | 1153 |
| $L_s$ | 92 | 92 |
| $C_s$ | 1024 | 1026 |

From these measurements, the simulation model is updated to estimate the various losses. The distribution of losses in the inverter is given in Figure 13 for the piezoelectric filtering and the passive LC filtering.

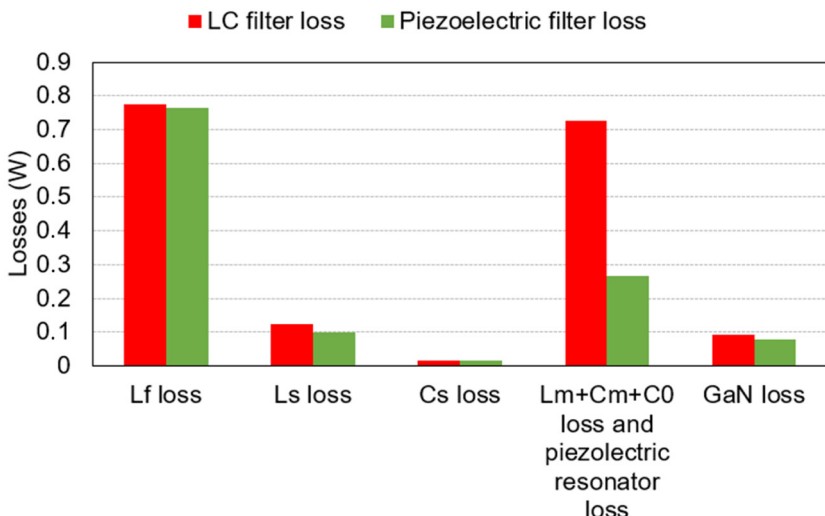

**Figure 13.** Distribution of losses in each prototype.

For both simulations (with piezoelectric filtering and LC filtering), we consider an input voltage of 30 V, an output power of 15 W, and a switching frequency of 10.4 MHz.

The total losses are 1.223 W (efficiency of 91.8%) with piezoelectric filtering and 1.733 W (efficiency of 88.4%) with LC filtering. There is a difference in losses compared to the experimental values (an efficiency of 88.1% for the piezo filter and 86.4% for the LC filter). This is because the simulation does not take into account all the losses, such as, for example, the parasitic inductances of the PCB or the integration of the piezoelectric resonator in the PCB. This difference can also be explained because the quality factors of the inductors are characterized by small signals, which is not the case during operation.

Figure 13 shows that a major part of the losses occurs in the inductance $L_f$ and the filter, whereas the output loop $L_s/C_s$ represents only 7% of the total losses. The losses in the GaN component are mainly conduction losses since the GaN HEMT switches to ZVS mode in the simulation.

The equivalent passive components ($L_m + C_m + C_0$) represents 725 mW against 266 mW for the piezoelectric resonator. Using a piezoelectric component decreases the total losses by 510 mW, i.e., a reduction of 30% in global losses.

## 6. Conclusions

This paper presents the operating principle and the design of a class L−Piezo inverter. The advantages of using a piezoelectric resonator to filter the second harmonic are detailed and demonstrated. A high−quality factor of the piezoelectric component is it significantly reduces the losses in the second harmonic filter. Furthermore, the piezoelectric filter adds stability in terms of frequency compared to L−C discrete components having a wide tolerance range. The gain in volume compared to the traditional LC branch, especially the air inductor, is also an interesting point for using a piezoelectric resonator (8.1 cm$^3$ for air inductance vs. 0.16 cm$^3$ for resonator). A prototype operating at 10.4 MHz for 30 V input voltage and 15 W validates the high performance of the piezoelectric filtering compared with a traditional L−C filter. Using a piezoelectric material reduces the losses in the second harmonic filter by 60%. Although it is more complicated to integrate, a piezoelectric resonator provides higher efficiency, higher power density, and reduces electromagnetic emissions compared to an inductor.

## 7. Patents

The class L−Piezo inverter topology presented in this document is a CEA patent (US11431264).

**Author Contributions:** Conceptualization, V.M.; methodology, V.M. and G.D.; validation, V.M., G.D., S.C. and X.M.; formal analysis, V.M. and G.D.; writing, V.M.; review, G.D. and S.C. All authors have read and agreed to the published version of the manuscript.

**Funding:** CEA−LETI is a Carnot institute. This research work was funded by the French ANR via Carnot funding (N° 19 CARN 0005 01).

**Conflicts of Interest:** The authors declare no conflict of interest.

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
