# Peer review of "Comparison between Piezoelectric Filter and Passive LC Filter in a Class L−Piezo Inverter"

_electronics, doi:10.3390/electronics11233983_

Round 1

Reviewer 1 Report

  The authors have presented a comparison between piezoelectric filter and passive LC filter in a class L-Piezo inverter. The idea of piezoelectric inverter is new and the paper is well written. However, the following modification should be considered in the manuscript.

-        The introduction section is poor. several recent similar works should be cited and explained in this section. Also, the related inverters, such as class E, Class EF2, Class EF3, and Class E2 should be explained and compared with the presented work. At the end of the introduction section, compare the proposed work with the cited works to show the advantages of the proposed work and its novelty. Also, emphasize the novelty of the proposed work in the abstract.

-        Some typos should be removed. For example, between values and unit a space must be exist, i.e. “10.4 MHz”, “15 W”.

-        The English level of  the manuscript should be improved.

-        The voltage waveforms (Vds, Vgs, Vout) of the prototype without piezoelectric material is not shown.

-        The output voltage of the L-Piezo inverter is not pure sinusoidal, which shows the nonlinearity of the proposed amplifier. Please explain the reasons or possible solutions.

Author Response

Hello,

Thank you for reviewing our article.

Sincerely.

Vincent Massavie

Reviewer 2 Report

In the paper “Comparison between piezoelectric filter and passive LC filter 2 in a class L-Piezo inverter” the authors present the advantage of piezoelectric filter in rapport with LC filter in a particular case. The advantage of the piezoelectric filter vs. the LC filter has been well known for almost a century, and it is probably not necessary to highlight this advantage in every particular case. 

 The presented paper is well designed in the theoretical presentation, in the simulation and in the experimental section but unfortunately it does not bring a significant difference (1.7%) in efficiency but can be retained as an alternative to the extent that the cost/efficiency ratio is satisfied.  

 The authors must review the introduction in order to provide readers with information about the importance of the field approached and, on this occasion, to present the main novelty elements brought by this study.

 The typographic space is not used efficiently, I recommend a compression of the figures in the format (a), (b) whenever possible such as for example in the case of Fig.1/2 Fig, 8/9, Fig. 11/12, Fig 13/14.

Author Response

(The authors gave the same response as above.)

Round 2

Reviewer 1 Report

The comments are addressed in the manuscript and the manuscript can be accepted after following modifications.

-        The references should be cited in the numerical order in the manuscript; for example [12]-[14] then [19] [20] should be corrected.

-        In addition to the circuit elements in Figure 7, some elements are added in the fabricated prototype; Kindly explain the elements applied in the fabricated circuit which are not mentioned in the manuscript.  

Author Response

References have been put in the correct order.

The circuit that appears in Figure 9(a) is a ZVS (Zero Voltage Switching) regulation that is the subject of another article. In this article, this circuit is not used and the inverter operates in open loop. An explanatory sentence was added line 282 and the reference of the article on the ZVS regulation was added.

Reviewer 2 Report

In the revision version of the paper the authors improve the introduction section and now the contribution in the context is more clear. The suggestion about graphical compression was done.  

Author Response

Thank you for your review.